# Implementation and staff perceptions of a quality assurance system in a Finnish private hospital during the COVID-19 Pandemic: A qualitative study

Mira Huusko[1], Anni Vuohijoki[2,3]*, Leena Ristolainen[4], Juhana Leppilahti[2], Sanna-Maria Kivivuori[5], Heikki Hurri[4]

1 Finnish Education Evaluation Centre, Helsinki, Finland, 2 Translational Medicine Research Unit, Medical Research Center Oulu, Oulu University Hospital, Oulu, Finland, 3 Orton Orthopaedic Hospital, Helsinki, Finland, 4 Research Institute Orton, Helsinki, Finland, 5 Helsinki University Hospital, Helsinki, Finland

☯ Mira Huusko and Anni Vuohijoki contributed equally to this work

* anni.vuohijoki@student.oulu.fi

## Abstract

### Background

Implementing quality assurance systems in healthcare settings is a growing trend globally. However, the impact of such systems on staff well-being and patient safety, particularly during major disruptions such as the COVID-19 pandemic, remains insufficiently explored. This study investigates how the implementation of the Joint Commission International (JCI) accreditation system affected a Finnish private hospital during a period of organisational change and crisis.

### Methods

This qualitative study was conducted through seven focus group interviews in two rounds (2020 and 2022) with a total of 27 healthcare professionals. The participants represented various professional groups in the hospital. Data were analysed using conventional qualitative content analysis to identify themes related to staff experiences, perceived benefits, and challenges associated with the quality assurance system.

### Results

The implementation of the quality assurance system was initially viewed positively, especially for its potential to unify guidelines, support staff orientation, and improve patient safety. However, the process was perceived as time-consuming and misaligned with local practices. The COVID-19 pandemic and changes in hospital ownership significantly slowed implementation. Despite this, the system was considered

**Data availability statement:** All anonymized data are available in the Supplementary materials. The original interview recordings are stored at the Research Institute of Orton.

**Funding:** This work was supported by Research Institute Orton through grants from the Ministry of Social Affairs and Health in Finland, grant no. A2500/495. There was no additional external funding received for this study. The funders had no role in study design, data collection and analysis, decision to publish, or preparation of the manuscript.

**Competing interests:** The authors have declared that no competing interests exist.

beneficial for institutional credibility, marketing, and internal clarity of roles and responsibilities.

## Conclusions

While comprehensive quality assurance systems such as JCI can enhance process transparency and patient outcomes, their successful integration requires contextual adaptation, dedicated resources, and sustained leadership. Staff well-being and participation should be prioritized throughout the process to ensure long-term benefits.

## Introduction

Accreditations have been part of healthcare for a long time [1]. Accreditation systems have become integral parts [2–4] partly due to globalisation and the growth of medical tourism [5]. International accreditations have emerged alongside national systems to enable the assessment of hospital quality across borders, focusing on standardising protocols to maintain efficiency and safety [6].

Systematic monitoring of work atmosphere and patient safety incidents is an essential component of any quality assurance system. Accredited systems, such as the Joint Commission International (JCI), are particularly emphasised. Because accreditation entails a large investment, evaluating its benefits is important. Some studies have investigated these benefits [7–9] but evaluating these accurately is difficult and studies are heterogenous. The quality assurance systems are widely used in healthcare and other sectors [10,11], but how their implementation affects staff well-being and patient safety remains underexplored [12–15]. It is reasonable to assume that well-being at work and work satisfaction are strongly correlated with good outcomes in healthcare service delivery, including patient safety [16–18]. Thus, examining both motivating and demotivating factors for staff is essential when studying the effects of quality assurance systems [12,19]. In line with the study's focus, this research emphasises staff experiences and the practical implementation of accreditation, rather than technical or regulatory details, to maintain clarity and relevance to the research questions.

In this study, we examine a particularly distinctive case: a Finnish private hospital specialising in orthopaedic surgery (joint replacements and back surgery) and rehabilitation. The hospital experienced an ownership transfer to a public hospital organisation during the COVID-19 pandemic, providing a unique context in which accreditation coincided with significant organisational change. These circumstances allow us to explore how accreditation interacts with staff well-being, patient safety, and organisational resilience under rarely studied conditions, highlighting the interplay between organisational change, crisis management, and quality initiatives.

Determining the effects of quality assurance systems is challenging due to multiple confounding factors, such as hospital size, patient health status, and management culture [20]. Accreditation and hospital quality system implementation are expensive and time-consuming, which can create legitimacy concerns for hospital management

[21,22]. Previous research has examined accreditation outcomes in various contexts [23–25]. However, little attention has been paid to staff attitudes during concurrent crises or in Nordic private hospital settings.

Work satisfaction comprises favourable or unfavourable feelings and emotions employees associate with their work [26]. Work satisfaction can be considered more subjective than well-being and is considered more subjective than well-being [27]. Well-being at work is a more comprehensive concept that includes job satisfaction, though the distinction is minimal [28,29].

Patient safety and employee well-being are correlated [30–34]. We investigate the effectiveness of hospital accreditation system regarding patient safety and employee well-being using a qualitative approach, to provide a comprehensive picture of the hospital during the introduction of the quality system. As Øvretveit and Gustafson [35] noted that accreditation costs can be minimised if implementation is efficient. Factors critical for success include management and physician engagement at all levels, robust data systems, appropriate training, and effective project team management [36]. By focusing on staff perceptions and experiences, this study directly addresses the research questions and ensures that background information supports understanding the rationale for examining these perceptions

This qualitative study explores hospital staff perceptions of the implementation of a quality assurance system, aiming to describe their experiences and views on the system established in a Finnish private hospital. It contributes to international literature by examining accreditation in a Nordic private hospital, which is underrepresented in previous research. The focus on an orthopaedic specialist hospital adds relevance, as surgical care and rehabilitation require standardised processes and present specific patient safety challenges. Moreover, the simultaneous ownership transition and the COVID-19 pandemic create a rare opportunity to explore accreditation under conditions of organisational change and crisis. Together, these factors allow the study to shed light on how accreditation influences staff attitudes, well-being, and perceptions of patient safety in a highly dynamic environment—a perspective largely absent from prior studies.

The research questions are as follows:

1. What are the expectations of the staff regarding the implementation of the quality assurance system?

2. What advantages and disadvantages do hospital staff perceive in the implementation of the quality assurance system?

## Materials and methods

This qualitative research was conducted through seven focus group interviews to determine the perceptions of different hospital staff towards the quality system and its implementation. The purpose of the qualitative interviews was to broaden personnel's baseline views, conceptions and expectations regarding quality assurance and their own well-being before the implementation of JCI standards.

The study was conducted in a Finnish private hospital specialising in orthopaedic surgery (joint replacements and back surgery) and rehabilitation. Data was collected during two periods: in 2020 and in 2022. These periods coincided with an ownership transfer and the COVID-19 pandemic, creating a context of organisational change and crisis management.

The interviewed focus groups comprised randomly assigned hospital staff. Private practitioners were also included in the study population, so the groups were multiprofessional. For the purposes of the interview groups, all employees of private hospital Orton were compiled into four lists: List 1 comprised physicians, List 2 nurses, List 3 therapists, and List 4 administration, support services, and secretarial staff. In addition, independent practitioners were identified and classified according to the same scheme. Within each list, individuals were arranged in alphabetical order and subsequently assigned a random number generated using Excel's built-in =RAND() function. The individuals were then reordered based on the random number, which determined their position in the selection order.

Because the aim was to assemble three discussion groups, the first three individuals from each randomised list were invited to participate. In cases of absence or refusal, replacements were drawn in order from the randomised list (starting

from the fourth position onward). Participation was voluntary, and each selected individual was given the opportunity to opt out. In total, 156 employees were included in the randomisation process out of a total staff of 196, corresponding to 80% of the personnel at the time. Hourly workers and individuals working less than one day per week were excluded, as were members of the research team. Independent practitioners were included.

Qualitative data were first collected in three thematic group interviews in 2020, and a total of 12 people participated. In 2022, four additional focus group interviews were conducted in which 15 people participated. Thus, a total of 27 employees were interviewed across seven focus groups.

In 2020, three of the twelve physicians (25%) participated, along with three nurses (7.7%), three therapists (13%), and three administrative or other staff members (4%). The 2020 groups included both full-time (9/12) and part-time (3/12) staff. In 2022, four physicians (30%) participated, with the nursing group representation remaining unchanged at 5.8%. Participation increased slightly among therapists (17%) and among administrative or other staff (6%). This composition is broadly reflective of the overall personnel distribution of the hospital at the time.

Across both years, participants represented a wide range of ages (approximately 30–70 years), professional seniority, and employment types (full-time, part-time, and independent practitioners). In 2020, eight women and four men participated, whereas in 2022 nine women and six men took part, resulting in a slightly more gender-balanced sample. Approximately half of the participants in 2022 (50%) had also taken part in the 2020 focus groups, allowing longitudinal comparison of changes in staff perceptions during the implementation of the quality system. The remaining participants were new, as the groups were complemented due to staff turnover between the two data collection points.

These interviews were conducted face-to-face, but one participant took part in one online interview. Each focus group comprised three to four participants representing different professional groups, ensuring that discussions were multidisciplinary and reflected a range of perspectives. At the beginning of each interview, participants were informed about the ethical principles of the study: participation was voluntary, they could withdraw at any time, discussions were confidential, the data would be anonymized, and they had the right to refuse permission for audio recording. The interviews lasted, on average, 51 minutes each, ranging from 34 to 60 minutes. They were conducted in Finnish at two different periods (2020 and 2022) because the research team wanted to understand the changes that had occurred in the participants' views on the quality assurance system. The interviews were performed by an external interviewer with knowledge of different quality assurance systems and qualitative research. All participants provided written informed consent for both participation in the study and audio recording of the interviews. The researcher also signed a confidentiality agreement to ensure the protection of participant privacy.

Only one of the participants attended the initial training on the JCI quality assurance system. The interviews were recorded with the permission of the participants and transcribed into a total of 162 pages of text.

## Data collection process

The data collection process consisted of the following components

- Participants were selected using randomised lists compiled from all hospital staff, including physicians, nurses, therapists, administrative or support staff, and independent practitioners. Hourly workers, those working less than one day per week, and members of the research team were excluded.

- Three to four participants were assigned to each focus group to ensure representation of different professional groups (multiprofessional composition).

- A total of seven focus group interviews were conducted: three in 2020 with 12 participants and four in 2022 with 15 participants. Approximately half of the 2022 participants had also participated in 2020, allowing for longitudinal comparison.

- Interviews were scheduled at times convenient for staff

- Interviews were conducted face-to-face, except for one online

- Discussions followed a semi-structured interview guide covering perceptions of the quality system, well-being at work and expectations

- Duration: Interviews lasted on average 51 minutes (range 34–60 minutes)

- Audio recording and transcription: All interviews were audio-recorded with participant consent and transcribed verbatim, generating 162 pages of text

- Observational notes were taken by the researcher to supplement verbal data

- Data analysis: Transcripts were analysed using conventional qualitative content analysis. Open-ended responses were coded using NVivo version (1.7), grouped into categories and higher-order themes, and then interpreted in relation to the quality system's implementation progress, advantages, and disadvantages.

- Ethical considerations: Written informed consent was obtained from all participants for both participation and audio recording. Confidentiality was ensured, and the researcher signed a confidentiality agreement. According to the institutional research ethics committee no ethical approval was needed (see S1 Checklist)

The interviews were analysed using conventional qualitative content analysis, which focuses on identifying, coding, and categorising patterns or themes that emerge directly from the textual data [37]. First, all interview transcripts were read multiple times to gain an overall understanding of the data. Next, the transcribed interviews were coded using NVivo. Codes were then grouped into categories and higher-order themes based on recurring patterns and frequency of occurrence. The aim of the analysis was to gain a comprehensive understanding of how the focus groups perceived the quality assurance system, its usefulness, and its development from the viewpoints of employees from different professional categories. To enhance the credibility and dependability of the findings, preliminary themes were discussed within the research team (peer debriefing), reflexive notes were maintained by the primary researcher to account for potential biases, and interpretations were reviewed iteratively. The most prevalent themes were then selected for reporting. Finally, the results were analysed in relation to the quality system's implementation progress, as well as its perceived advantages and disadvantages.

## Results

### Initial excitement and optimism in 2020

According to the interviewees, the implementation of the quality system started well. The process of building the JCI quality system progressed by conducting briefings and training, setting up working groups, collecting compiled data and updating guidelines in both 2020 and 2022. During the COVID-19 pandemic, JCI meetings were held online. A dedicated page on the hospital intranet was developed to keep track of the progress of the quality system and updated guidelines.

The interviewees revealed that the implementation of the quality assurance system later slowed down because of the lack of a full-time project manager. In 2020, many interviewees acknowledged that they had little information on how a quality assurance system should be implemented.

### Perceived benefits of the quality system

Interviewees recognised that there could be many benefits of the implementation of the quality assurance system. In particular, the compilation of guidelines and principles was considered a major possible improvement. The quality assurance system could help improve the various processes in the hospital and enhance operations. According to the interviewees, the system helps promote patient safety, customer service and communication within the hospital and between staff.

'I believe it has a really strong, significant impact on patient safety. (...) I think it brings things like… I mean, there are already checklists and things like that in use, but with this, you can really make sure that everything that needs to be

considered has been covered. I really believe in that. And if everyone knows the criteria, what's been done and what needs to be done, then it should work. That's how I see it. So yeah, it definitely relates to safety.' (Interview 1, 2020)

'In my opinion, our patient safety is already so good that even a 99% improvement would be very marginal—hardly a significant change. Maybe a single mistake could be prevented, but that's really why I was talking earlier about going through and developing the processes, which is the kind of work that should be done regardless of any larger umbrella certification.' (Interview 2, 2022)

Interviewees felt that, ideally, a quality system would facilitate their own work, other than patient work, as well as the work of hospital managers. The quality system and the compilation of common guidelines could also be useful for the induction of new employees.

'And then if a new person joins, a student or someone else, he/she can be taught, given instructions … this is how we do it in [hospital] … you can lean on something a little bit in the orientation'. (Interview 1, 2020)

In addition, the systematic collection of feedback and the increased efficiency of the operating room were considered positive outcomes. The development of the quality assurance system could also help gain patients' confidence.

'I don't know. At first, I found it difficult years, years ago—no help, always feedback—but it's very good [now]. We're here for the patients, the clients, the rehabilitators.' (Interview 1, 2020)

'Well, of course, it's kind of lonely work, you know. But yeah, good manners and making sure the patient feels they can trust you – that's important. And of course, there's the responsibility too.' (Interview 1, 2020)

'I'm sure it gives a positive image to the outside world, like, we have quality standards and quality control in place, so to me, that really shows that we care about what we do. And yeah, you often hear that patients have done their research on our place and staff thoroughly before they even come here.' (Interview 1, 2020)

The interviewees saw a link between the quality assurance system and their own well-being and motivation at work. Common guidelines and written responsibilities make their work clearer and increase their motivation, safety and well-being at work.

'Clarity and I think there will be some kind of effect on the work atmosphere because everyone knows how it's done. Then you don't have to say or think that someone failed to do it again.' (Interview 1, 2020)

According to the interviewees, a clear quality assurance system would allow errors and quality shortcomings to be detected as early as possible, thereby reducing them in the long run. Administrative work and processes were thought to improve with the quality system, which also helped in the marketing efforts of the hospital.

'It sounds like this could serve a marketing purpose. [It's] good to compete with other companies selling similar services [and emphasise] that we already have this kind of quality control since 2020 or 2021 … it gives a certain advantage'. (Interview 2, 2020)

## Perceived challenges and issues

A disadvantage identified by the interviewees was that setting up a quality assurance system required much work. It was time consuming because of the rigour required for accreditation, and implementing it was not easy, especially when starting a demanding JCI accreditation process. Many interviewees felt that the system was disconnected from their own work

and that it would make the hospital's work more rigid, if all rules and guidelines are followed to the letter. The fear was that if working time was spent on implementing of the quality system, there would not be enough time for patients and patient care.

'Now that I'm talking, I realise that I'm approaching it in a way that it's brought in from the outside, without it having any relevance to my work at the moment'. (Interview 2, 2020)

'Well, I mean, if you had to follow everything exactly by the book, it just wouldn't work. If we did everything exactly the way, it's written down—like how we're supposed to operate and how things are audited—there's no way to follow it to the letter. So yeah, it kind of feels like it could get a bit too rigid, and hopefully, it won't end up being too strict.' (Interview 3, 2020)

'Yes, it's just that whether there is a risk that there will be this kind of rigid activity and extra, which may not want to spend time on this patient, I do not know.' (Interview 3, 2020)

'I don't know, maybe if it becomes a mechanical action, then there is a disadvantage. Close contact with the patient should not be affected.' (Interview 3, 2020)

According to the interviewees, the introduction of the quality assurance system did not reduce their actual workload, as the amount of patient work would remain the same, despite the potential benefits. They also raised concerns that because the quality assurance system was developed externally, it was not well suited to the Finnish hospital environment.

### Impact of external variables in 2022

When the next interviews place in 2022, the implementation of the quality assurance system had not progressed as much as originally assumed. The reasons for this were the COVID-19 pandemic, the change of ownership of the hospital a few years earlier and the consequent increase in the number of patients. The new owner had abandoned the JCI quality assurance system and was considering moving to an ISO-based system [2,4] (2) (4). According to the interviewees, switching to another quality system would not be much of a disadvantage. They felt that the work done would not be wasted and could instead be used to set up a new quality assurance system. However, the fact that some of the people responsible had moved on to other tasks slowed down the implementation process.

'Then the group leader wanted to move away from it, so then I moved away, too. Somehow it went like that.' (Interview 1, 2020)

'This work won't go to waste, as the ISO standard has the same aspects; it's just a simplified version. So, let's focus more on the most relevant parts. Apparently, the decision whether to include JCI at all has still not been 100% made'. (Interview 3, 2022)

As in the 2020 interviews, the staff expressed in the 2022 interviews that the quality assurance system provided the greatest benefits in terms of producing and compiling common guidelines. It also helped with the induction of new staff and the improvement of different hospital processes and patient safety. The interviewees explained how the existence of quality standards is useful, as these would help the hospital in various tenders and are part of the functioning of a modern hospital.

'Major efforts were made to mark everything down. For example, I'm working at a different placement, and now I work during Saturdays at the ward, [and] it has been very easy—based on having read the data myself first and then having supervised a few times—so now I can fill in if someone is absent. From this, you can see that it [the quality assurance system] has probably had an effect'. (Interview 2, 2022)

'Well, I guess it's the modern time, that you must have. […] Well, I guess it has to do with the bidding and stuff with the municipalities. You work with them, and they have their own definitions, their own standards that the applicant must meet. I'd guess you need to have some kind of quality standard just to be part of the competition in the first place. So, it's just one of those things that a company this size must have.' (Interview 3, 2022)

In the 2022 interviews, the respondents acknowledged the link between quality management and well-being at work. The quality assurance system was also thought to help in patient safety, marketing the hospital, with patients choosing the hospital as their place of treatment.

'If you think about it this way, that this quality standard makes us more equal professionally, and this is the thing we all do in this profession … we're all equal in our profession, then it's personal [when someone] … feels that something is a burden. But the fact [is] that in this workplace and profession, things are done in a specific way, bringing boundaries and security to work, so everyone's reactions to things are personal, but there are always supporting workplace guidelines. Having something to compare your work could bring about well-being'. (Interview 3, 2022)

'When you think about being abroad for a longer period and getting sick, and then you must choose the hospital. I would go to one that has met some quality standard and, either JCI or ISO system, rather than another that has not. Before I know anything about those hospitals.' (Interview 2, 2022)

'Patients know that we monitor ourselves and we have a quality system or monitoring in our work, so it has a marketing value to know what is being done.' (Interview 1, 2022)

'From a practical perspective and in terms of patient safety, there is indeed a lot of potential here—many opportunities and benefits. In fact, I would almost hope that this would continue. So, some kind of quality system where we could go through the entire system, and then perhaps even receive some kind of certification, which could also be used in marketing, for example… and it certainly wouldn't hurt.' (Interview 4, 2022)

The interviewees found it useful to record activities and tacit information and to improve the transparency of all activities. They also recognised some link between the quality system and their own work motivation and work satisfaction.

'It brings transparency to how things are done and how they are. Of course, every workplace has its own ways of doing things, but the fact [is] that quality comes from standards and professional ethics … [which] are written down. So, it's also doing things the way they are meant to be done. (Interview 3, 2022)

'It's like, when the employer shows that our work is valued, it also shows in how the work environment is kept pleasant. We all have a part in making our work environment enjoyable. It's about us being happy and... that affects things too, but also the facilities need to be decent. That's part of the quality of the job and adds to how good the work environment feels.' (Interview 2, 2022)

'It's easy to work here. The vibe is open. There's no kind of hierarchy where you're afraid to ask someone anything, no matter what their title is in the company. And if that stays, I think that's when things really work. It's that team spirit, and that's what I'm used to here, and it's what keeps everything going. I think that's a big part of job well-being.' (Interview 2, 2022)

## Evolution of perspectives over time in 2022

However, the interviewees faced significant challenges in implementing the accreditation system. They found the selected JCI quality system to be overly complex and detailed for effective use in a Finnish hospital. According to the interviewees,

the JCI system was too complicated and time-consuming, as it was primarily designed for American hospitals and insurance companies. Furthermore, the hospital was experiencing financial difficulties during that period.

'It kind of started off... how should I put it? Way too complicated, when really it should've been more straightforward. Back when we were going through the manual about how this should be done, it gave the impression that this was a system coming from an American environment, where you'd need to figure out how the electricity coming from the socket is even generated. Quality started with things like that, so it was hard to see any relevance in the requirements from our perspective. We had one or two meetings, and somehow, we ended up talking about the bottlenecks in the operating room that have been discussed for the past 30 years in this place and will probably be talked about for the next 30 years. We identified some issues.' (Interview 2, 2022)

'The only thing I know about ISO is that it's lighter—maybe, in some ways, a reduced version of JCI. It's more focused on European and international hospitals, while JCI is not so much [on that], as it's more focused on preventing insurance companies from tricking hospitals. If they notice that a hospital has breached some JCI guidelines, they'll pull out of insurance claims, and then there are long litigations. JCI is just more focused on that side of things. The ISO standard is a sure thing, but I don't know [much about it] yet, as the decision hasn't been made'. (Interview 3, 2022)

'[It] feels superimposed. If we start thinking about my job description and how to improve my tasks, [I] don't see that the accreditation system really has anything to do with it'. (Interview 1, 2022)

According to the interviewees, the COVID-19 period and the change of ownership of the hospital ended the enthusiasm for building a quality assurance system. In 2022, there was uncertainty among the interviewees about how the quality assurance system could move forward.

'When COVID-19 happened, the JCI project had been going on for about half a year. We then switched to online meetings, and it sort of got forgotten … all things related to it, [including] our enthusiasm. But I think some groups have updated the old guidelines and started writing down the new ones as well'. (Interview 3, 2022)

## Discussion

According to the interviewees, the results of this study showed that the introduction of an accredited hospital quality system could help improve the hospital's common guidelines and processes, including those on the induction of new employees. The implementation of the accredited quality system could also be linked to the hospital's work atmosphere and well-being at work and work motivation.

Hospital staff linked the accreditation process to hospital management, the operational culture, common goals and the use of the data collected. According to the interviewees, the accreditation system could help highlight the good care services provided by the hospital and reinforced its good reputation. The pursuit of accreditation was linked to the hospital's identity, image and marketing to attract new customers [38].

The results suggest that accreditation serves both instrumental and symbolic purposes. Instrumentally, the system provided a structured framework to compile guidelines, improve processes, support patient safety, and facilitate staff orientation. Symbolically, accreditation enhanced the hospital's reputation and marketing appeal, contributing to its identity and external image. This dual role raises critical questions: when does accreditation risk becoming more of a symbol than a practical tool? Several interviewees indicated that the burden of accreditation sometimes overshadowed its practical benefits, suggesting a potential symbolic emphasis:

'Now that I'm talking, I realise that I'm approaching it in a way that it's brought in from the outside, without it having any relevance to my work at the moment'. (Interview 2, 2020)

This aligns with literature indicating that externally imposed quality certifications may emphasise image management and marketing over meaningful clinical improvement [39,40].

The symbolic focus of accreditation can have unintended consequences for staff morale and well-being. Interviewees reported feelings of stress, increased workload, and a perception that their clinical work was not directly supported by accreditation activities. This highlights the importance of balancing symbolic recognition with practical utility to maintain engagement and motivation.

'If we did everything exactly the way, it's written down (...) there's no way to follow it to the letter. So yeah, it kind of feels like it could get a bit too rigid, and hopefully, it won't end up being too strict.' (Interview 3, 2020)

These findings support previous studies suggesting that accreditation can be perceived as an external imposition if it does not align with staff workflows or local contexts [36].

The enthusiasm for achieving the accreditation ended during the COVID-19 pandemic, the change of ownership of the hospital and other organisational changes a few years earlier. At the same time, Finland began to face a severe shortage of skilled nursing staff and financial challenges for the hospital, the healthcare system and the state economy. Hospital staff was at that point too busy to set up an accreditation system and perceived it as a burden for them.

The key elements in establishing an accreditation of quality assurance systems include clear goals, committed management, systematic processes, the collection of information to support operational improvements, an organisational culture that fosters accreditation and quality assurance and a willingness to enhance practices [41]. Implementation must consider contextual readiness, including financial and organisational stability, and staff availability to engage meaningfully in quality assurance processes [38].

Quality assurance systems are needed in healthcare. Quality assurance is a continuous, internal process aimed at ensuring that services and processes meet specified standards. Accreditation focuses on maintaining and improving standards of care as well, but it is a formal, external validation process by an independent accrediting body requiring a rigorous review, including documentation, site visits and periodic reassessments. This study critically emphasizes that if accreditation is pursued mainly for external recognition or marketing, its instrumental value for improving safety, efficiency, or staff well-being may be limited.

However, the added value of accreditation is difficult to examine using a comparative study design for practical and ethical reasons [42]. When pursuing major initiatives, such as accreditation, hospitals need to evaluate the corresponding outcomes. With an efficient quality assurance system, employees experience increased feelings of coherence, leading to improved well-being at work [43]. Externally accredited quality has an impact on a hospital's brand and marketing, which in turn may cause extra burden for the employees. These insights suggest that the implementation of accreditation should balance symbolic and instrumental objectives. Hospitals should consider carefully whether accreditation initiatives are genuinely aimed at improving patient safety, operational efficiency, and well-being at work, or if they are primarily pursued for external recognition. Future research could explore the long-term consequences of symbolic versus instrumental accreditation, and how different implementation strategies affect staff engagement, well-being, and organisational resilience. The qualitative findings of this study provide a basis for hypothesis generation and suggest that more nuanced, context-sensitive approaches to accreditation are needed in healthcare systems with stringent regulatory environments. This study contributes to a better understanding of how accreditation should be adapted to the specific circumstances of different hospitals in various countries, where regulation and legislation may vary significantly.

There are many potential confounding factors in this field of research, as healthcare is culture specific and strictly regulated by law. The strength of qualitative research is that it brings out perspectives that have not been integrated into the research questions. The qualitative approach to research is exploratory and helps develop hypotheses for future research [37].

## Strength & limitations

The study offers valuable insights into the establishment of a quality system during the COVID-19 pandemic. The findings highlight the challenges faced by a private hospital as it encounters multiple changes simultaneously. Since the study uses qualitative interview data, it provides in-depth, context-specific insights into the experiences, perceptions, and challenges of healthcare professionals and administrators during the implementation of the quality system at two phases (2020 and 2022). This approach uncovers nuances that quantitative data might overlook, offering a deeper understanding of the interdependencies and complexities involved.

As the study focuses on a private hospital during the COVID-19 pandemic, its findings cannot be directly generalized to public healthcare systems or other environments where resources, patient populations, and operational frameworks differ significantly. Although the qualitative approach provides rich and nuanced insights, it is susceptible to potential bias in data collection and interpretation. The personal perspectives of healthcare staff and hospital administration may not fully represent the broader hospital community, which limits the scope of the findings.

The study was conducted during the COVID-19 pandemic, a period characterised by exceptional circumstances, which may not reflect normal operations. The challenges faced and the solutions implemented may therefore be unique to the pandemic context, thus limiting the applicability of the findings to standard hospital operations.

The study's sample size is limited, particularly since it focuses on a single hospital and a small group of individuals. This limitation may affect the diversity of perspectives and weaken the generalizability of the findings to other organisations. The study does not account for the long-term effects of the implemented quality assurance system, as the rapid implementation during the pandemic does not allow for a thorough evaluation of the system's long-term impact on hospital operations. Long-term follow-up would be necessary to assess both the benefits and potential drawbacks of the quality system.

## Conclusion

In conclusion, establishing an accredited quality assurance system requires substantial effort and investment raising the question of whether the benefits outweigh the costs. Accreditation initiatives should be adapted to the specific context of the hospital. This includes considering the existing regulatory framework, available resources, and organisational structure, ensuring that the system complements rather than duplicates existing practices. Existing studies indicate that accreditation may be particularly valuable in countries with less robust regulatory frameworks, in settings with sufficient funding and resources, and in areas where national guidelines are lacking or inadequate. In Finland, national-level supervision is already well established and thorough, reducing the marginal benefit of accreditation.

Accreditation occupies a complex position between an instrumental mechanism for improving care and a symbolic marker of legitimacy. While it can standardise processes, enhance safety culture, and promote accountability, its impact often depends on how deeply it is integrated into everyday clinical practice. When accreditation becomes primarily a means of demonstrating compliance rather than facilitating genuine improvement, it risks serving as a symbolic ritual rather than a functional tool. This dynamic reflects a broader tension within health systems, where external accountability mechanisms may incentivise form over substance. The resulting administrative burden can, in turn, erode professional autonomy and contribute to staff fatigue, especially when improvement efforts are perceived as top-down mandates disconnected from clinical realities. Sustaining morale therefore requires leadership that frames accreditation as developmental, not punitive, and ensures that evaluation processes produce tangible learning rather than bureaucratic repetition.

At a systems level, the symbolic weight of accreditation also intersects with growing cross-border mobility in healthcare. Within the European Union, regulatory frameworks enable patients to seek elective care in other member states, introducing both opportunities and vulnerabilities. Accreditation may offer a reassuring signal of safety and quality across jurisdictions, yet this signal can obscure substantial differences in clinical outcomes, follow-up arrangements, and

liability standards. In a market where patients increasingly shop for care, the over-reliance on accreditation as a proxy for quality risks commodifying healthcare and amplifying inequities between those able to navigate such systems and those who cannot. A more balanced approach would couple accreditation with transparent, outcome-based data and continuity-of-care safeguards, ensuring that legitimacy derives from demonstrable standards of practice rather than the mere possession of a certificate.

Strong managerial commitment is essential for successful implementation. The study showed that even when staff were generally positive about the quality system, achieving accreditation would have required more dedicated leadership, clear goals, and consistent support from management. Managers play a central role in allocating resources, facilitating processes, and fostering a culture that values quality improvement.

Supporting well-being at work and work satisfaction is critical. Ensuring that employees feel valued, engaged, and equipped with the necessary training and resources contributes directly to the effectiveness of any quality assurance system. A positive work environment, collaborative culture, and open communication between leadership and staff not only sustain morale but also enhance the overall performance and sustainability of quality initiatives.

These findings suggest that the successful implementation of accreditation depends on balancing contextual fit, managerial commitment, and employee engagement. Future research should examine how these factors interact to influence the long-term outcomes of accreditation in healthcare settings with varying regulatory, financial, and organisational conditions

In Finland, national-level supervision is well established and thorough. In countries where such oversight is absent, accreditation plays a crucial role in standardizing hospital operations. While staff generally had a positive attitude towards quality systems accreditation in the present study, even stronger managerial commitment and significant resources would have been needed to achieve the accreditation. Ensuring quality despite limited resources remains a key challenge, requiring effective strategies and innovative approaches.

Supporting well-being at work and work satisfaction is essential for the success of any quality assurance system whether accredited or not. A positive work environment, where staff feel valued and engaged, can contribute to higher levels of motivation and improved performance. Fostering a culture of collaboration, recognition, and open communication between leadership and staff is key to sustaining staff morale and ensuring the long-term success in implementing quality initiatives. Ensuring that staff have the necessary support, training, and resources to carry out quality initiatives is crucial for both their professional satisfaction and the overall success of the hospital.

## Supporting information

**S1 Checklist. Human Participants Research Checklist.**
(DOCX)

**S2 Dataset. Anonymized interview transcripts.**
(PDF)

## Author contributions

**Supervision:** Leena Ristolainen, Juhana Leppilahti, Sanna-Maria Kivivuori, Heikki Hurri.

**Writing – original draft:** Mira Huusko, Anni Vuohijoki.

**Writing – review & editing:** Leena Ristolainen, Heikki Hurri.

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
