## [Decision Letter · Decision Letter 0]

31 Aug 2025

Dear Dr. Vuohijoki,

Thank you for submitting your manuscript to PLOS ONE. After careful consideration, we feel that it has merit but does not fully meet PLOS ONE’s publication criteria as it currently stands. Therefore, we invite you to submit a revised version of the manuscript that addresses the points raised during the review process.

We look forward to receiving your revised manuscript.

Kind regards,

Elif Ulutaş Deniz

Academic Editor

PLOS ONE

Journal Requirements:

This work was supported by the Orton Research Institute through grants from the Ministry of Social

Affairs and Health in Finland, grant no. A2500/495. There was no additional external funding received for this study.

3. Thank you for stating the following in your Competing Interests section: No

6. Please remove all personal information, ensure that the data shared are in accordance with participant consent, and re-upload a fully anonymized data set.

Additional guidance on preparing raw data for publication can be found in our Data Policy (https://journals.plos.org/plosone/s/data-availability#loc-human-research-participant-data-and-other-sensitive-data) and in the following article: http://www.bmj.com/content/340/bmj.c181.long .

Reviewers' comments:

Reviewer's Responses to Questions

**Comments to the Author**

1. Is the manuscript technically sound, and do the data support the conclusions?

Reviewer #1: Yes

Reviewer #2: Yes

2. Has the statistical analysis been performed appropriately and rigorously?

Reviewer #1: Yes

Reviewer #2: N/A

3. Have the authors made all data underlying the findings in their manuscript fully available?

Reviewer #1: Yes

Reviewer #2: No

4. Is the manuscript presented in an intelligible fashion and written in standard English?

Reviewer #1: Yes

Reviewer #2: Yes

Reviewer #1: Implementation and Staff Perceptions of a Quality Assurance System in a Finnish Private

1. Abstract:

• The phrase "impact on staff well-being and patient safety" is used, however the abstract doesn't go into detail on staff well-being later on. This produces a gap between the purpose and the findings. So, please make the research goal and the results more similar. You can add results about well-being to the findings.

2. Introduction: The introduction is thorough and well-organised. It talks about the development of accreditation, the gaps in previous research, and the specific goal of the study. Nevertheless, certain amendments are required such as:

• Line 57: "Accreditations have been a part of traditional industries..." Could be reworded to immediately focus on healthcare instead of broad industries.

• Consider compose a concise critical synthesis paragraph clarifying that previous research has examined accreditation outcomes in various situations; however, none have focused on staff attitudes during an associated crisis involving COVID-19 and ownership transfer within a Nordic private hospital environment. This research addresses such gap.

• line 82: "This situation brings up an important question: Is accreditation worth pursuing…?” This is a strong rhetorical question; however, it would be better if it were tied directly to the research goal instead of being left being hanged.

• Line 88, The text calls the difference between work satisfaction and well-being "negligible," nevertheless it is important. This could make it harder for readers to understand. Suggestion: This study asserts that well-being and satisfaction are interrelated dimensions of staff experience.

• The novelty is not distinctly indicated; the introduction resembles a conventional discourse on accreditation rather than providing a compelling rationale for this specific investigation. Through the incorporation of the key characteristics (Finnish private hospital, orthopaedic specialist, ownership transition during COVID-19) are just emphasised at the conclusion and appear to lack significance.

3. Materials and Methods: this section is clear and thorough; however, certain modifications are needed such as:

• Make it clear whether these practitioners work full-time, part-time, or on a temporary basis, as this has a big effect on how people see them.

• Readers can't tell how representative the sample is because there are no percentage distributions for staff groups including doctors, nurses, physiotherapists, and administrative staff.

• The two rounds of data collection conducted in 2020 and 2022 are referenced but lack clear explanation. Were the same individuals involved in the follow-up, or were different groups utilised?

• Include additional information regarding the coding process, triangulation, and the measures implemented to ensure rigour, such as audit trails, reflexivity, and peer debriefing.

4. Result: an extensive dataset, authentic staff perspectives, and a distinctive long-term outlook. However, some amendments are needed such as:

• The current section resembles an introduction rather than an analytical discourse. There are themes, for example: initial optimism, practical benefits, staff well-being, rigidity and workload, contextual constraints, ownership shift, and pandemic disruption; however, they lack clear differentiation It is advisable to split the content into four to six main concepts, accompanied by subheadings. For example:

Initial excitement and optimism (2020)

Perceived benefits include guidelines, orientation, patient safety, and marketing.

Perceived issues include excessive effort, overly strict regulations, and lack of connection to the local context.

Impact of external variables, such as COVID-19, ownership transitions, and staff turnover

Evolution of perspectives over time (reflecting on 2022, progressing towards ISO)

5. Discussion: Balanced and contextual, acknowledging both advantages and disadvantages; however, modifications are necessary such as:

• Strengthen the critique by comparing the instrumental and symbolic aims of accreditation, substantiated by citations. as the section says that accreditation may be more useful for marketing than for patient safety, however this might be critically expanded:

Does certification run the risk of becoming more of a symbol than a tool?

How does this stress effect the morale of the staff?

6. Conclusion:

• The comparison between Arab countries and Finland is presented suddenly, without a grounding in the study's data or supporting sources. It could look like guesswork instead than facts that have been proven. It is prudent to undertake this comparison with enhanced clarity for example, "Existing studies demonstrate that in nations with less robust regulatory frameworks..."

• The conclusion talks about a number of things, such as resources, health, culture, teamwork, and training. Even though all the parts are important, the message seems a little unclear. Think about focussing on two or three main suggestions: making accreditation fit the situation, making sure managers are committed, and using employee satisfaction as a measure of performance.

Reviewer #2: This is an excellent piece of work addressing an important and relatively unexplored area of study. The topic is timely and relevant, particularly given its focus on the implementation and staff perceptions of a quality assurance system in a Finnish private hospital during the COVID-19 pandemic. Below are my detailed comments and suggestions for improvement.

Background

=As the opening chapter, the background requires important revisions. It should serve as the gateway to the problem under study and must remain focused on the research title. Some paragraphs appear tangential and not directly relevant to the topic; I recommend revising these to maintain focus.

= Notably, COVID-19 is not mentioned at all in the background, which is surprising given its centrality to the research. The pandemic context should be explicitly integrated to strengthen the rationale for the study.

= A synthesis of existing literature should be included to demonstrate familiarity with prior work and to clearly establish the research gap.

= The final paragraph of the background (“The research setting is a Finnish private hospital, however owned by a large public hospital...”) would be more appropriately placed under the Methodology section, particularly in the subsection describing the study setting.

Materials and Methods

= The discussion of data collection techniques and analysis approaches is clear and well-articulated.

= However, the methods section would be stronger if it began with the study design and study setting, to provide the reader with a clear orientation before delving into methodological details.

= The description of focus group discussions (FGDs) needs clarification:

o How many FGDs were conducted, and what was the exact procedure?

o Who participated? Were all 12 people plus the additional 16 part of the same groups, or different ones? Please clarify

how the total of 28 participants was organized.

= Provide more detail on participant selection, including criteria, demographic characteristics, and any ethical considerations applied.

= Include a step-by-step description of the data collection process (e.g., how interviews were conducted, whether observations were used, duration, recording, transcription, etc.).

= Expand on the data analysis process: how was the qualitative data processed, coded, and interpreted? Specify the analytical framework or technique applied (e.g., thematic analysis, grounded theory).

= Outline the strategies used to ensure quality and credibility (e.g., triangulation, member checking, reflexivity).

= Provide a clear account of the ethical procedures followed, including consent processes and institutional approval, if applicable.

Results, Discussion, and Conclusion

= These sections are generally well-written and coherent.

= However, I suggest that the strengths and limitations of the study be presented immediately after the discussion. This is a common academic practice and allows the reader to critically assess the scope and validity of the findings.

Overall Impression

The manuscript is strong and promising, but it requires revisions to improve clarity, focus, and methodological rigor. Addressing the above points will enhance the credibility, scholarly value, and overall impact of the study.

**Do you want your identity to be public for this peer review?** For information about this choice, including consent withdrawal, please see our Privacy Policy

Reviewer #1: No

Reviewer #2: No

---

## [Author Response · Author response to Decision Letter 1]

16 Oct 2025

Please note that we have attached the entire revised manuscript as an appendix, as the number of modifications was considerable.

Remarks for the Editor

We have carefully reviewed the provided guidelines and have made every effort to revise the document in accordance with your instructions.

This work was supported by the Orton Research Institute through grants from the Ministry of Social

Affairs and Health in Finland, grant no. A2500/495. There was no additional external funding received for this study. Please state what role the funders took in the study. If the funders had no role, please state: "The funders had no role in study design, data collection and analysis, decision to publish, or preparation of the manuscript."

Thank you for you comment. We have updated our Funding statement at the row 71 as follows:

This work was supported by the Research Institute Orton through grants from the Ministry of Social Affairs and Health in Finland, grant no. A2500/495. There was no additional external funding received for this study.

We have added "The funders had no role in study design, data collection and analysis, decision to publish, or preparation of the manuscript." to our Cover letter.

3. Thank you for stating the following in your Competing Interests section: No

Thank you for your comment and the information change to the online submission form. We have added:

"The authors have declared that no competing interests exist.” to our cover letter

Thank you for your clarification regarding data collection. We have now added the complete anonymized dataset to the Supplementary Materials 3.

5. Please include captions for your Supporting Information files at the end of your manuscript, and update any in-text citations to match accordingly. Please see our Supporting Information guidelines for more information: http://journals.plos.org/plosone/s/supporting-information

Thank you for your comment. We have added a “Supplementary Materials” section after the reference list and included corresponding references to it within the main text.

6. Please remove all personal information, ensure that the data shared are in accordance with participant consent, and re-upload a fully anonymized data set.

This has been completed. A fully anonymized dataset has been reviewed for compliance with participant consent and added to Supplementary Materials 3.

Thank you for your comments. We have addressed the reviewer’s feedback in the “Comments to the Author” section.

Review Comments to the Author

1. Abstract:

• The phrase "impact on staff well-being and patient safety" is used, however the abstract doesn't go into detail on staff well-being later on. This produces a gap between the purpose and the findings. So, please make the research goal and the results more similar. You can add results about well-being to the findings.

Based on your comments, we have extensively revised the manuscript. The specific modifications are detailed in the following sections.

2. Introduction: The introduction is thorough and well-organised. It talks about the development of accreditation, the gaps in previous research, and the specific goal of the study. Nevertheless, certain amendments are required such as:

• Line 57: "Accreditations have been a part of traditional industries..." Could be reworded to immediately focus on healthcare instead of broad industries.

We sincerely thank you for your valuable feedback. In response, we have revised the sentences on line 57 to the following form:

Accreditations have been part of healthcare for a long time [1]. Accreditation systems have become integral parts [2-4] partly due to globalisation and the growth of medical tourism [5]. International accreditations have emerged alongside national systems to enable the assessment of hospital quality across borders, focusing on standardising protocols to maintain efficiency and safety [6].

• line 82: "This situation brings up an important question: Is accreditation worth pursuing…?” This is a strong rhetorical question; however, it would be better if it were tied directly to the research goal instead of being left hanging.

Thank you for your kind comment. We have removed the question from line 82. Instead, we have now expanded the Discussion section accordingly.

• Line 88, The text calls the difference between work satisfaction and well-being "negligible," nevertheless it is important. This could make it harder for readers to understand. Suggestion: This study asserts that well-being and satisfaction are interrelated dimensions of staff experience.

Thank you for your thoughtful comment. The revision can be seen in the manuscript starting from line 11. We have modified the sentence to the following form:

Well-being at work is a more comprehensive concept that includes job satisfaction, though the distinction is minimal [25, 26]

• The novelty is not distinctly indicated; the introduction resembles a conventional discourse on accreditation rather than providing a compelling rationale for this specific investigation. Through the incorporation of the key characteristics (Finnish private hospital, orthopaedic specialist, ownership transition during COVID-19) are just emphasised at the conclusion and appear to lack significance.

Thank you very much for this insightful comment. We have thoroughly revised the latter part of the Introduction to highlight the novelty and context of our study more clearly. The modifications can be found starting from line 116.

Patient safety and employee well-being are correlated [30-34]. We investigate the effectiveness of hospital accreditation system regarding patient safety and employee well-being using a qualitative approach, to provide a comprehensive picture of the hospital during the introduction of the quality system. As Øvretveit and Gustafson [35] noted that accreditation costs can be minimised if implementation is efficient. Factors critical for success include management and physician engagement at all levels, robust data systems, appropriate training, and effective project team management [33]. By focusing on staff perceptions and experiences, this study directly addresses the research questions and ensures that all background information supports understanding the rationale for examining these perceptions

This qualitative study explores hospital staff perceptions of the implementation of a quality assurance system, aiming to describe their experiences and views on the system established in a Finnish private hospital. It contributes to international literature by examining accreditation in a Nordic private hospital, which is underrepresented in previous research. The focus on an orthopaedic specialist hospital adds relevance, as surgical care and rehabilitation require standardised processes and present specific patient safety challenges. Moreover, the simultaneous ownership transition and the COVID-19 pandemic create a rare opportunity to explore accreditation under conditions of organisational change and crisis. Together, these factors allow the study to shed light on how accreditation influences staff attitudes, well-being, and perceptions of patient safety in a highly dynamic environment—a perspective largely absent from prior studies.

The research questions are as follows:

What are the expectations of the staff regarding the implementation of the quality assurance system?

What advantages and disadvantages do hospital staff perceive in the implementation of the quality assurance system?

3. Materials and Methods: this section is clear and thorough; however, certain modifications are needed such as:

• Make it clear whether these practitioners work full-time, part-time, or on a temporary basis, as this has a big effect on how people see them.

• Readers can't tell how representative the sample is because there are no percentage distributions for staff groups including doctors, nurses, physiotherapists, and administrative staff.

• The two rounds of data collection conducted in 2020 and 2022 are referenced but lack clear explanation. Were the same individuals involved in the follow-up, or were different groups utilised?

Thank you for your valuable comment. The Materials and Methods section has undergone substantial revisions. To save space, we have not included the entire modified section here. The updated version of Section 2, Materials and Methods, can be found starting from line 157.

4. Result: an extensive dataset, authentic staff perspectives, and a distinctive long-term outlook. However, some amendments are needed such as:

• The current section resembles an introduction rather than an analytical discourse. There are themes, for example: initial optimism, practical benefits, staff well-being, rigidity and workload, contextual constraints, ownership shift, and pandemic disruption; however, they lack clear differentiation. It is advisable to split the content into four to six main concepts, accompanied by subheadings. For example:

Initial excitement and optimism (2020)

Perceived benefits include guidelines, orientation, patient safety, and marketing.

Perceived issues include excessive effort, overly strict regulations, and lack of connection to the local context.

Impact of external variables, such as COVID-19, ownership transitions, and staff turnover

Evolution of perspectives over time (reflecting on 2022, progressing towards ISO)

Thank you for your helpful suggestions regarding the Results section. We have added the following subheadings to improve clarity and structure:

Initial excitement and optimism in 2020 — line 285

Perceived benefits of the quality system — line 295

Perceived challenges and issues — line 344

Impact of external variables in 2022 — line 372

Evolution of perspectives over time in 2022 — line 442

In addition to these changes, we have also added content related to patient safety starting from line 421.

‘From a practical perspective and in terms of patient safety, there is indeed a lot of potential here—many opportunities and benefits. In fact, I would almost hope that this would continue. So, some kind of quality system where we could go through the entire system, and then perhaps even receive some kind of certification, which could also be used in marketing, for example… and it certainly wouldn’t hurt.’ (Interview 4, 2022)

5. Discussion: Balanced and contextual, acknowledging both advantages and disadvantages; however, modifications are necessary such as:

• Strengthen the critique by comparing the instrumental and symbolic aims of accreditation, substantiated by citations. as the section says that accreditation may be more useful for marketing than for patient safety, however this might be critically expanded:

Does certification run the risk of becoming more of a symbol than a tool?

How does this stress effect the morale of the staff?

Thank you for these thoughtful and thought-provoking follow-up questions. We have thoroughly revised the entire Discussion section to address these points in greater depth. To save space, the full updated text can be found starting from line 506.

6. Conclusion:

• The comparison between Arab countries and Finland is presented suddenly, without a grounding in the study's data or supporting sources. It could look like guesswork instead than facts that have been proven. It is prudent to undertake this comparison with enhanced clarity for example, "Existing studies demonstrate that in nations with less robust regulatory frameworks..."

Thank you for your comment. We have removed the entire paragraph starting from line 679.

• The conclusion talks about a number of things, such as resources, health, culture, teamwork, and training. Even though all the parts are important, the message seems a little unclear. Think about focussing on two or three main suggestions: making accreditation fit the situation, making sure managers are committed, and using employee satisfaction as a measure of performance.

Thank you for your valuable comments. Starting from line 688, we have added the following content to the Conclusion section:

Strong managerial commitment is essential for successful implementation. The study showed that even when staff were generally positive about the quality system, achieving accreditation would have required more dedicated leadership, clear goals, and consistent support from management. Managers play a central role in allocating resources, facilitating processes, and fostering a culture that values quality improvement.

Supporting staff well-being and work satisfaction is critical. Ensuring that employees feel valued, engaged, and equipped with the necessary training and resources contributes directly to the effectiveness of any quality assurance system. A positive work environment, collaborative culture, and open communication between leadership and staff not only sustain morale but also enhance the overall performance and sustainability of quality initiatives.

Taken together, these findings suggest that the successful implementation of accreditation depends on balancing contextual fit, managerial commitment, and employee engagement. Future research should examine how these factors interact to influence the long-term outcomes of accreditation in healthcare settings with varying regulatory, financial, and organizational conditions

In Finland, national-level supervision is well established and thorough. In countries where such oversight is absent, accreditation plays a crucial role in standardizing hospital operations. While staff generally had a positive attitude towards quality systems accreditation in the present study, even stronger

---

## [Decision Letter · Decision Letter 1]

2 Nov 2025

Implementation and Staff Perceptions of a Quality Assurance System in a Finnish Private Hospital During the COVID-19 Pandemic: A Qualitative Study

PONE-D-25-27467R1

Dear Dr. Vuohijoki,

We’re pleased to inform you that your manuscript has been judged scientifically suitable for publication and will be formally accepted for publication once it meets all outstanding technical requirements.

Kind regards,

Elif Ulutaş Deniz

Academic Editor

PLOS ONE

Reviewers' comments:

Reviewer's Responses to Questions

**Comments to the Author**

Reviewer #1: All comments have been addressed

Reviewer #2: All comments have been addressed

2. Is the manuscript technically sound, and do the data support the conclusions?

Reviewer #1: Yes

Reviewer #2: Yes

3. Has the statistical analysis been performed appropriately and rigorously?

Reviewer #1: Yes

Reviewer #2: Yes

4. Have the authors made all data underlying the findings in their manuscript fully available?

Reviewer #1: Yes

Reviewer #2: Yes

5. Is the manuscript presented in an intelligible fashion and written in standard English?

Reviewer #1: Yes

Reviewer #2: Yes

Reviewer #1: Each and every one of the comments has been handled, and the authors have made some outstanding changes.

Reviewer #2: = All the reviewers’ comments and suggestions have been thoroughly addressed, and the manuscript has shown significant improvement. I believe it now meets the required publication standards. Therefore, I recommend acceptance of the manuscript.

**Do you want your identity to be public for this peer review?** For information about this choice, including consent withdrawal, please see our Privacy Policy

Reviewer #1: No

Reviewer #2: No

---

## [Editor Report · Acceptance letter]

PONE-D-25-27467R1

PLOS ONE

Dear Dr. Vuohijoki,

I'm pleased to inform you that your manuscript has been deemed suitable for publication in PLOS ONE. Congratulations! Your manuscript is now being handed over to our production team.

Kind regards,

on behalf of

Dr. Elif Ulutaş Deniz

Academic Editor

PLOS ONE